# Spatial and Temporal Analysis of COVID-19 in the Elderly Living in Residential Care Homes in Portugal

**DOI:** 10.3390/ijerph19105921

**Published:** 2022-05-13

**Authors:** Felipa De Mello-Sampayo

**Affiliations:** Business Research Unit (BRU-IUL), Lisbon University Institute (ISCTE-IUL), 1649-026 Lisbon, Portugal; fdmso@iscte-iul.pt

**Keywords:** COVID-9, elderly, in residential care homes (RCH), spatial-temporal analysis, kernel density estimation (KDE), geographic weighted regression (GWR)

## Abstract

Background: The goal of this study is to identify geographic areas for priority actions in order to control COVID-19 among the elderly living in Residential Care Homes (RCH). We also describe the evolution of COVID-19 in RHC throughout the 278 municipalities of continental Portugal between March and December 2020. Methods: A spatial population analysis of positive COVID-19 cases reported by the Portuguese National Health Service (NHS) among the elderly living in RCH. The data are for COVID-19 testing, symptomatic status, comorbidities, and income level by municipalities. COVID-19 measures at the municipality level are the proportion of positive cases of elderly living in RCH, positive cases per elderly living in RCH, symptomatic to asymptomatic ratio, and the share of comorbidities cases. Spatial analysis used the Kernel density estimation (KDE), space-time statistic Scan, and geographic weighted regression (GWR) to detect and analyze clusters of infected elderly. Results: Between 3 March and 31 December 2020, the high-risk primary cluster was located in the regions of Braganca, Guarda, Vila Real, and Viseu, in the Northwest of Portugal (relative risk = 3.67), between 30 September and 13 December 2020. The priority geographic areas for attention and intervention for elderly living in care homes are the regions in the Northeast of Portugal, and around the large cities, Lisbon and Porto, which had high risk clusters. The relative risk of infection was spatially not stationary and generally positively affected by both comorbidities and low-income. Conclusion: The regions with a population with high comorbidities and low income are a priority for action in order to control COVID-19 in the elderly living in RCH. The results suggest improving both income and health levels in the southwest of Portugal, in the environs of large cities, such as Lisbon and Porto, and in the northwest of Portugal to mitigate the spread of COVID-19.

## 1. Introduction

COVID-19 has affected the elderly in residential care homes (RCH) internationally, but detailed information on it is scarce [1]. This article fills some of this gap in the literature. Internationally, institutional care settings for older adults have seen high rates of COVID-19 [2]. We analyze the evolution of COVID-19 in residential care homes in all 278 municipalities in continental Portugal. In one US home care [3], 33% of residents died due to COVID-19. In the US, 56% of home care residents testing positive for COVID-19 were asymptomatic, they had the symptoms four days after testing [4]. In the analysis of four care homes with COVID-19 outbreaks in London, UK, around 40% of residents tested positive, of whom 43% were asymptomatic at the time of testing and 18% had atypical symptoms [5,6]. The effect of COVID-19 has been substantial in RCH with known outbreaks. Shielding nursing home residents from potential sources of SARS-CoV-2 infection, and ensuring rapid action to minimize the extent of the outbreak if the infection is introduced, will be important for any subsequent wave [7].

Income and comorbidities may be relevant for the decision makers of the municipalities since people with fewer financial resources and poor health can be a hotspot for infection [8,9,10,11,12,13]. The income per capita is likely to capture broadly defined socio-economic factors characterizing municipalities. To prevent the spread of COVID-19, one needs to use gel alcohol hand sanitizer, individual masks, and water for cleaning in general [14]. Higher income is closely related to better living conditions, i.e., basic sanitation and access to personal and home hygiene products. Thus, the higher the municipalities’ income per capita, the lower the infection rate. People in low income or poverty levels are associated with low levels of health [15,16,17] and comorbidities and are thus, more vulnerable to the pandemic threat [11,12,13].

In 2020 the total population of continental Portugal was around 10 million, 22.5% of which were elderly and approximately 29.8% of the elderly live in residential care homes. Our study includes 14,053 elderly living in RCH that tested positive in 278 municipalities in continental Portugal, between 3 March 2020 and 31 December 2020. The elderly living in RCH account for 40.4% of the positive tests of the elderly population, and 6.8% of the total positive cases in Portugal. One of the earliest cases of the elderly living in residences in Portugal was identified in Vila Nova de Famalicao, Braga, in the northwest littoral of Portugal. Soon thereafter several cases appeared in Braga. On 19 March 2020, there were 19 cases in Vila Nova de Famalicao. This highlights the wide infection spread within the community in close municipalities. COVID-19 is a public health problem in which human mobility among municipalities is one of the main factors in the spatial distribution of the pandemic [14,18,19]. Spatial analysis identified priority areas of aging adults that may require unique palliative care in light of the COVID-19 pandemic [7,20]. Other studies have reported the efficacy of spatial interpolation for studies on the effects of variables on COVID-19 [14,21,22,23].

Several studies in public health and epidemiology [24,25,26,27], used Kernel Density Estimation (KDE). Kernel density estimation to analyze spatial patterns of point events. Lai et al. [28] identified quantitively unusual spread patterns of SARS outbreaks in Hong Kong, using KDE. This study also uses KDE to evaluate the spatial distribution pattern of elderly COVID-19 positive cases at the municipality level.

## 2. Method

A set of spatial analysis techniques were used to address the spatial and temporal analysis of COVID-19 in the elderly living in residential care homes in Portugal. Data from two sources, which are described below, were added together in order to examine the relationships between relative municipality income and the number of confirmed COVID-19 cases.

Our analysis is of public health data at the municipality level. The data are taken from the Business Intelligence of the information technology support platform for the National Epidemiological Surveillance System (BI SINAVE). We obtained data on the confirmed cases for COVID-19 from 3 March 2020 until 31 December 2020 for the following variables: the unique patient identifier, the date when the person was identified as a confirmed case of infection by SARS-CoV-2/COVID-19, the patient’s age when the case was confirmed (i.e., the date of notification), the municipality of Portugal where the confirmed-case patient was residing, and information on symptoms and signs collected at the time of notification: Asymptomatic or Symptomatic. Regarding the health level of the confirmed cases, the data also indicated if the patient had comorbidities.

We used ≥65 years to define “older people”, as this is considered the conventional retirement age [29]. Users of long-term care are on average 80 years or older, especially in institutional care, and are often women [29]. Since data on individuals were not available for the elderly living in residential care homes, we proxy it by elderly ≥80 years old. Four COVID-19 measures were created in order to analyze the spatial distribution of the positive cases of elderly living in RCH. First, the proportion of positive cases of elderly living in RCH relative to the positive cases of the public, in general, is calculated as the number of elderly positive cases divided by the total positive cases of a given municipality multiplied by 1000. Second, we calculated the percentage of positive, i.e., positive cases per elderly living in RCH, obtained as the number of positive tests divided by the total population in elderly residential care homes of a given municipality multiplied by 1000. This measure gives the positive cases per capita living in RCH. Third, we calculated the symptomatic to asymptomatic ratio as the number of symptomatic positive cases divided by the asymptomatic number of cases performed for a given municipality for the elderly living in the RCH. Finally, we included the share of comorbidities cases, calculated as the number of positive cases with comorbidities divided by the total number of positive cases for a given municipality for the elderly living in the RCH multiplied by 1000. The Income variable was measured by the expenditures of social protection benefits per active person (Old age and survivors) in euros for 2020 by the municipality from INE [30].

### 2.1. Kernel Density Estimation (KDE)

KDE is a non-parametric method that uses local information defined by kernels to estimate densities of specified features at given locations. The Kernel map indicates the areas with the highest and lowest density. The areas with higher densities of infection (red shades) on the map showed where there is the greatest probability of positive cases, whereas low density areas (blue shades) are those where there is a low probability of spreading COVID-19. To analyze local spatial autocorrelation, the Moran’s I [31] test statistic was calculated.

Given the infectious nature of COVID-19, cases have spread over large distances in space with time. COVID-19 point location data themselves do not provide information about change over time, nor do they provide insights into how it spread. In order to visualize the spread of COVID-19 from 3 March to 31 December 2020, KDE was also computed for each month. Thus, we created 10 kernel maps to characterize changes in the positive proportion of hot spots over 10 months in a chronologic sequence.

### 2.2. Space-Time Scanning

Spatiotemporal data may consist of not only spatial dependencies but also temporal as well as spatiotemporal dependencies [32]. Kulldorff [33] designed space-time scan statistic which enables detecting clusters in a given geographical region during a predetermined time-period. The space-time permutation scan statistic [34], captures purely spatial as well as purely temporal variations. It also addresses the problem of capturing the changes due to spatio-temporal interactions, by taking stratified random permutation.

Under the hypothesis that the municipalities with the highest positive rates of elderly with COVID-19 would form space-time clusters, data were analyzed by the space-time scanning technique. The Kulldorff Scan method [35] recognizes the spatial cluster that most likely violates the null hypothesis of non-agglomeration; so, it identifies the primary and secondary clusters. The SaTScan™ software (Kulldorff and Information Management Services, Inc., Boston, MA, USA) was used for spatial, temporal, and spatio-temporal analyses. Then, we used STATA (Stata Press. StataCorp., College Station, TX, USA, 2015) to visualize the relative risk in high-risk cluster areas.

The primary cluster is the maximum likelihood window, which has the lowest probability of occurring at random. A *p*-value is assigned to this cluster. For purely spatial and space-time analyses, SaTScan also identifies secondary clusters in the data, in addition to the most likely cluster, and sorts them according to Likelihood Ratio Analysis. The Poisson probabilistic model [36] was used to analyze the elderly at risk. We had the daily positive cases of the elderly living in residences and the total positive tests of each location within continental Portugal. The space-time scan is made in a round or elliptical window, the base of which is spatial and height refers to time.

In addition to identifying spatial clusters, the Kulldorff Scan gives a measure of the presence or absence of the event, or relative risk (RR). A value that represents how much an area is more or less likely to have an event in relation to the other areas of the entire territorial extension studied is calculated. RR indicates the incidence of positive cases of elderly living in residences in the cluster in relation to the incidence outside the cluster. To assess significance at *p* < 0.01, the Monte Carlo simulation was replicated 999 times.

Based on the assumption that there was a significant difference in the occurrence of positive cases for COVID-19 during 2020, data were analyzed according to this period with daily precision. The Poisson probability model was used for statistic scanning with daily precision and retrospective analysis, assuming a maximum of 25% of the population under risk in each cluster, and a maximum 80 km radius for the cluster size.

### 2.3. Geographically Weighted Regression (GWR)

This study uses Geographically Weighted Regression (GWR) [37,38] to analyze how distance to nearby municipalities affects RR. Under GWR, each observation is weighted according to its location. The spatial weight is based on the distance between a municipality and surrounding municipalities. In health literature [39,40,41], to proxy for the negative impact of travel between patients and healthcare providers, distance has been used because of its availability:(1)$$RR_{rj}=\alpha_{0j}\left(v_{j},y_{j}\right)+\alpha_{kj}\left(v_{j},y_{j}\right)\;{\left(X_{rj}\right)}_{k}+\varepsilon_{j},$$
where $RR_{rj}$ is the dependent variable: Relative risk’s point r at municipal j, $\alpha_{0}$ is the intercept; $X_{rj}$ is the matrix of the independent variables comprising income per capita, and comorbidity; $\alpha_{k}$ is the vector of estimated parameters; $\varepsilon_{rj}$ is the error term assumed i.i.d. Standardized variables; $\left(v_{j},y_{j}\right)$ represents the coordinate location of the data point *j*, in our case, municipals. Each observation is weighted according to its proximity to *j*. The weights used are the distance. STATA (Stata Press. StataCorp., College Station, TX, USA, 2015) was used to estimate Equation (1) and visualize the local coefficients.

A spatially adaptive weighting function was used since it considers different bandwidths according to the density of the data. In this function, the number of nearest municipalities is fixed so that the distance adjusts up to a maximum in order to find that number of municipalities. The caveat of this method is that the relatively arbitrary selection of bandwidth may over or under-smooth the original data. In this study, different parameters for bandwidth were tested in a sensitivity analysis [26,42]. The maximum distance is determined by testing a number of distances as well as choosing a limit based on behavior theory regarding healthcare accessibility [43].

The statistical inference of the GWR was conducted by Monte Carlo simulation. First, it was tested if the local model describes the data significantly better than a global model. Second, the spatial variation of the parameter estimates was tested. The Monte Carlo simulation resulted in an experimental distribution. Finally, an experimental significance level was obtained for the spatial variability of each individual parameter.

## 3. Results

In the 278 municipalities of continental Portugal, 207,838 positive cases of COVID-19 were reported in 2020. Total positive cases in Portugal and in RCH for each month are presented in Table 1. The elderly positive cases living in residences (EC) accounted for approximately 7% (14,081 cases) of the total positive cases (TC), i.e., EC/TC. Using the number of positive cases of elderly living in residential care homes per 1000 total positive cases, i.e., the proportion of positive (PP), PP = (EC/TC) × 1000, it represented around 10% (20,041 cases) of the positive cases, i.e., PP/TC. During this period the maximum number of Portuguese positive cases was 3796 on 3 November, and the maximum number of elderly living in RCH was 191 on 29 November. In relative terms, the maximum number of the elderly infected was 195 per 1000 positive cases on 10 May 2020. Daily positive cases and a positive proportion of the elderly living in residences plotted in Figure 1 show an increased concentration of COVID-19 occurrences from the end of October until the end of the year. Pairwise comparisons between elderly positive cases and the proportion of positive cases show the crossover points of the two measures in Figure 1 which may indicate the beginning of substantial disease spread across Portugal. Therefore, the time from 3 March through 15 May was the period of highest risk of COVID-19 infection among the elderly living in RCH, whereas the period from 13 October through 31 December marked the outbreak throughout the territory.

Figure 2 summarizes COVID-19 hotspots in Portugal considering cumulative COVID-19 measures from 3 March through 31 December 2020. The positive and significant Moran’s I coefficients indicate that similar values tend to cluster together, which confirms the geospatial clustering and thus infectious nature of the disease.

The spatial kernel density distribution of measures of confirmed COVID-19 cases in the municipalities of continental Portugal indicates that the population located in northwestern Portugal was at a higher risk of contracting COVID-19 (Figure 2a). The largest concentration of municipalities that confirmed elderly positive cases of COVID-19 as a proportion of total positive cases is located in the districts of Lisbon, Porto, Braga, and Viana do Castelo; some districts in the interior of Portugal show warm colors (red and orange). With the spatial analysis of confirmed cases of COVID-19 across continental Portugal, it is possible to note a decrease in the concentration of municipalities with confirmed positive cases, as indicated by the colors blue to green. These areas are mainly concentrated in the region of Alentejo, in the south of Portugal. There was a gradual spread from one district to another.

Figure 2b shows the spatial kernel density distribution of positive cases per elderly living in residential health care. It is similar to Figure 2a, but we see that the green areas become more evident. This may indicate that there are likely more people with coronavirus in the RCH who have not been tested yet and are potentially asymptomatic. Figure 2c shows the estimated kernel density distribution of the share of comorbidities. When comparing Figure 2a,b with the Figure 2c, there are considerable similarities, which suggests that the positive cases of elderly living in residences have comorbidities. Finally, Figure 2d shows the estimated kernel density distribution of the symptomatic to asymptomatic ratio, and it suggests that hotspot areas of symptomatic cases are closely related to the two measures of positive cases (Figure 2a,b) and with the share of comorbidities positive cases (Figure 2d).

One of the goals of this research is to understand the spatial extent to which the pattern of COVID-19 cases changed during the period under analysis among the elderly living in the RCH. COVID-19 point location data alone do not provide information about change over time, nor do they provide insights into how it spread. In order to visualize the spread of COVID-19 from 3 March to 31 December 2020, KDE was also computed for each month.

We also analyzed a series of kernel maps based on monthly data on the proportion of positive cases of COVID-19 in the elderly living in RCH. There were disease hotspots in the north of Portugal in 2020. This was followed by a dissipation of incidence between June and August when the hotspots were isolated in smaller areas. This trend culminated in August when the hypothesis that the data are randomly distributed is not rejected. By September, new hotspot areas emerged in central, northeast, and southeast Portugal, and become larger between October and December. There is clear evidence of varying degrees of clustering as the epidemic evolved, with higher degrees of clustering occurring around the peak of the infection, first between March and April, and second, between November and December, and relatively small divergences from a random distribution between May and September.

In Table 2 space-time clusters of COVID-19 positive cases of elderly living in residential care homes and respective RRs, 2020 are listed. All clusters are statistically significant, and the RR varies between 1.55 and 5.14. The primary high-risk cluster was located in the municipalities in the north of Portugal, comprising the regions of Bragança, Guarda, Vila Real, and Viseu, with an RR of 3.67, between 30 September and 3 December 2020. This Cluster is shown in Figure 3 by the green municipalities. We also see an increase in both RR and clusters’ extension in the West and Northwest regions of Portugal, as well as in the Southwest (cluster light brown in Figure 3). Lisbon and Porto had the highest RR, the turquoise and navy blue clusters, respectively, in Figure 3.

In terms of timing, the first cluster occurred between 24 March and 13 April 2020, in the northwest of Portugal, shown in Figure 3 by the municipalities in pink. Then, it spread to Aveiro, Castelo Branco, Coimbra, and Guarda (brown cluster), Porto (yellow cluster), and Viseu (navy blue cluster). At the end of September, five clusters emerged, where the green and red clusters appear as those with the highest number of cases of the elderly living in residences. The green cluster was the longest lasting, with 75 days, followed by the red cluster with 67 days (Table 2).

Table 3 presents the results of the GWR model. Table 3 is arranged into two main sections, the first is composed of the first column which corresponds to the ordinary least squares (OLS). The rest of the columns correspond to the GWR model. The “Si” column reports the spatial variability test, the “Mean” column, reports the mean of the local estimates, and the other two columns show the maximum (Max) and minimum (Min) of the local estimates. The results of the global model (OLS) are as expected, yet the spatial variation is significant, suggesting that they do not characterize the local relationships between RR and income and comorbidity. Moreover, the Moran’s I value calculated from the residuals of the global model was positive and significant, indicating that the model suffered from spatial autocorrelation.

The mean, minimum, and maximum GWR coefficients are shown in Table 3. As shown in Figure 4, the spatial distribution of the parameters indicates the degree of spatial non-stationarity and illustrates the interesting way in which the effects vary over space. The spatial variation in the income per capita parameter depicts the differing effect of income per elderly across Portugal. It had a negative effect in 47% of the municipalities. Contrary to the global models, the mean of the Comorbidity parameter is negative, while remaining positive in 56% of local models.

## 4. Discussion

Analyzing COVID-19′s spread pattern is fundamental in guiding the next steps toward overcoming the harmful effects on the elderly living in residential care homes. This paper analyzes the confirmed cases of COVID-19 spatially among the elderly living in residential care homes across 278 municipalities of continental Portugal during 2020. Although officially published data regarding RCH are only partially available in European countries, according to the report of the London School of Economics, data from varying official sources show that nearly half of all COVID-19 deaths appear to be occurring in nursing homes, as in several European countries [44]. In Portugal, RCH provides 24-h nursing care and residential care for older adults who cannot be accommodated at home or in other settings, due to increasing frailty and care-related needs [45,46]. Data for the residential care home population are scarce and data sources are fragmented, meaning that our understanding of the needs and outcomes of residents is poor [1,20]. This article helps to fill this gap in the literature.

To analyze the spatial distribution of COVID-19 among the elderly living in residential care homes, we used the proportion of positive cases of elderly living in RCH and the percent positive. The percent positive measure must be interpreted with caution because the numerator (positive cases) is conditional on having been tested, whereas the denominator (total elderly population) is unconditional. It is a proxy of the percent positive of elderly living in the residences and may indicate that there are likely more people with coronavirus in the RCH who have not yet been tested. The proportion of positives is a critical measure because it gives us an indication of how widespread the infection is among the elderly living in residences, where the positive tests are occurring, and whether levels of disease transmission in RCH are the same as levels of disease transmission everywhere in Portugal. The proportion of positives will be high if the number of positive tests is too high, or if the number of the total population testing positive is too low. We also used the symptomatic to asymptomatic ratio and the share of comorbidities cases to indicate areas of greatest risk of spread of COVID-19.

Mapping of disease has become an essential public health instrument, along with the use of spatial analysis techniques to help in planning, managing problems, and in aiding decision making in the spread of epidemics [47,48]. To explore the spatial distribution pattern of COVID-19 cases at the municipality level we applied Kernel Density Estimation (KDE). We estimate the Kernel density for each COVID-19 measure, suggesting that the positive cases of elderly living in RCH have comorbidities and are symptomatic [4,5,49,50]. A high density of cases occurs mainly in the Northwest of Portugal and around the metropolitan regions (red color) of Lisbon, Porto, Braga, Viseu, and Coimbra and the concentration of cases gradually diminishes in the surrounding municipalities. The lower concentration of cases in the countryside and the municipalities that are more disconnected from the metropolitan regions may account for the low infection rates in the south of Portugal. These results are in accordance with Laires et al. [11], who suggest that the North of mainland Portugal had 72.8% of selected morbidity. Furthermore, the northern Portuguese region had a mainly elderly population. Older people are at the highest risk of COVID-19. The Lisbon and Tagus Valley region and Central Portugal region were the regions that also had large populations at risk for COVID-19.

Since we also seek to understand the spatial extent to which the pattern of COVID-19 cases changed during 2020 among the elderly living in the RCH, the space-time scan technique was employed, aiming to identify spatial and space-time clusters and their statistical significance. In studies specifically on COVID-19 [22], the authors used space-time scanning to identify areas of priority for disease control. The relative risk (RR), varies between 1.55 and 5.14. From March to December 2020 the municipalities of Porto and Viseu had clusters with more than twice the risk (RR = 5.14) of the clusters of Lisbon (RR = 1.83) and Porto (RR = 1.58), two other municipalities with a high risk of COVID-19 infection. The primary high-risk cluster was located in the municipalities in the north of Portugal comprising the regions of Bragança, Guarda, Vila Real, and Viseu, with an RR of 3.67, between 30 September and 3 December 2020. One may also see an increase in both RR and clusters’ extension in the West and Northwest regions of Portugal, as well as in the Southwest. Lisbon and Porto had the highest RR. In terms of timing, the first cluster occurred between 24 March and 13 April 2020, in the northwest of Portugal. Then, it spread to Aveiro, Castelo Branco, Coimbra, and Guarda, Porto, and Viseu. At the end of September, five clusters emerged with the highest number of cases of the elderly living in RCH. The longest-lasting cluster had 75.

In the present study, we find an interposition between the clusters of risk for positive cases of COVID-19 with low income and comorbidities. Social, economic, and health vulnerabilities may make the elderly more exposed and lead them to suffer more from COVID-19, resulting in higher rates of infection in nearby areas, especially when they do not have adequate access to healthcare [9]. The spatial distribution of the GWR parameters indicates the degree of spatial non-stationarity and illustrates the interesting way in which the effects vary over space. The global OLS model masked the local relationships between the dependent variables and the correlates. The spatial variation in the income per capita parameter had a negative effect in 47% of the municipalities. This result corroborates the literature in which income is likely to capture broadly-defined socio-economic factors characterizing health in each municipality. The epidemiology literature reports that the level of income is shown to negatively affect the spread of disease; wealthier individuals are able to enjoy better hygiene and living conditions, which is more difficult for low income persons [9,14,22,51]. Furthermore, people with low income or poverty are associated with low levels of health [15,16,17], and a negative relationship between municipalities’ income per capita, and relative risk of transmission is expected to occur. Accordingly, *ceteris paribus*, a low relative risk of transmission in richer municipalities is observed.

The local GWR analysis may shed light on the reasons for the mixed results of income and comorbidities regarding the transmission rate in each municipality. The spatial non-stationarity indicates a positive association between income per capita and a negative relationship between comorbidities with positive cases in 53% and 44% of the municipalities, respectively. A possible explanation for these counterintuitive effects might be the human mobility effect captured by the weight matrix of the GWR model. The weights used are the distance between one municipality and surrounding municipalities within a certain radius. Our use of the Kernel function to measure municipality spatial accessibility is gradually discounted as distance increases. Thus, human mobility and/or contact with staff that were asymptomatic dominated the income and comorbidity effects in some municipalities. In Wuhan, China, human mobility was the main factor affecting the spatial distribution of COVID-19 positive cases [18]. When human mobility was controlled, its effect decreased substantially and the infected population ceased to increase or even decreased in some locations. Rex et al. [14] and Chen et al. [19] report that population mobility was the main source of infection. Additionally, these authors suggest seeking to understand risk trends in different regions in order to ensure preparedness at the individual and organizational levels and prevent further outbreaks.

Some limitations to this analysis should be mentioned. Portugal had a low testing rate in the population in 2020, and there may be a different pattern than the one presented here. The paucity of studies carried out in relation to the spatial distribution of COVID-19 in the elderly living in residences makes it difficult to perform the proper comparisons. Nonetheless, some considerations are important in the interpretation of the findings. First, the information presented corresponds to the information available at the date of the notification, and not to the information resulting from the course of the disease. Laboratory reports do not contain information about the clinical presentation or the context of the infection, and variables on hospitalization and death are not presented. National data are aggregated, which means our understanding of the variation between residential care homes is limited. In order to better describe the evolution of outbreaks of COVID-19 in residences in continental Portugal, specifically the timing of outbreaks, number of confirmed cases, and relation with comorbidities and income, additional detailed information on how COVID-19 has affected the elderly in residential care homes is needed.

This work can provide valuable information to support government monitoring and predict the dissemination of viruses among the elderly community. Elderly persons living in residential care homes with comorbidities and low income are especially vulnerable to COVID-19, and upon contracting it are likely to die or to have lasting side effects [2,3,4,5,6,7,9,11]. Therefore, it is very important that health services are readily available to RCH and that control measures be applied to avoid the human mobility source of infection. Testing and tracing are crucial to an effective COVID-19 response, and care homes should keep a temporary record of current and previous residents, visitors and staff. The digital literacy of older people needs to be enhanced so they can use mobile apps for receiving information and for communicating with family members and community service providers even when physically separated. The World Health Organization demanded governments step up mitigation measures to control the disease’s spread to save lives [52]. The research and development of methods to detect and monitor the disease are necessary, but depend mainly on data support [53]. The spatial-temporal analysis is fundamental to monitoring the pandemic development and then applying mitigation measures accordingly.

## 5. Conclusions

This study spatially analyzed the positive cases of COVID-19 in the elderly living in RCH across 278 Portuguese municipalities in continental Portugal. It associates local and temporal data to indicate the areas with a continued risk of spread in RCH. We identified areas of high risk of infection to the elderly due to COVID-19 in the continent of Portugal in 2020. The results suggest improving both income and health levels in the southwest of Portugal, in the environs of large cities, such as Lisbon and Porto, and in the northwest of Portugal to mitigate the spread of COVID-19. This research opens new horizons for the development of future studies with more robust databases with respect to the elderly living in residential care homes.

## Figures and Tables

**Figure 1 ijerph-19-05921-f001:**
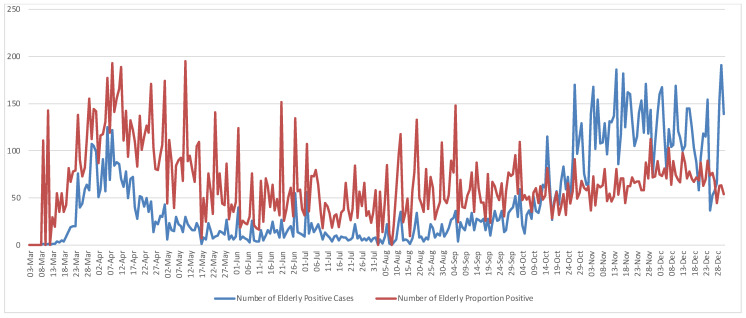
Daily Number of Elderly Positive Cases and Number Proportion Positive.

**Figure 2 ijerph-19-05921-f002:**
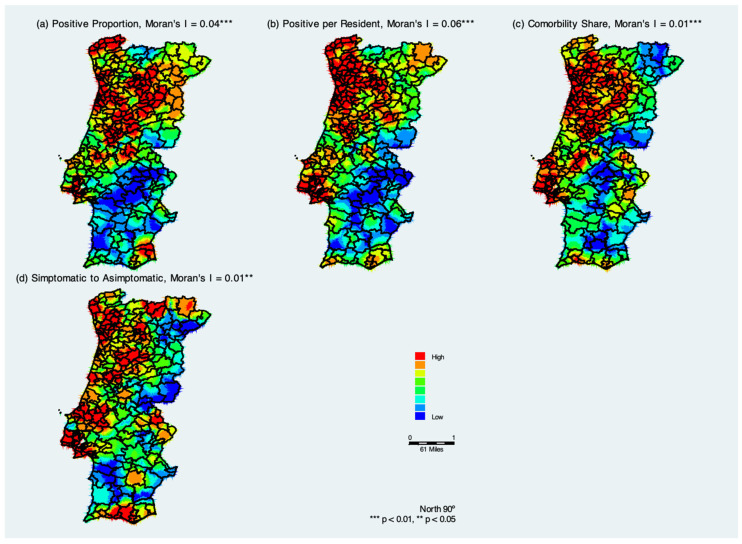
Kernel density estimation of COVID-19 infection 4 measures in residential care homes Portugal based on cumulative positive cases from March through December 2020. Subfigure (**a**) Elderly positive cases as a proportion of total positive cases; (**b**) Positive cases per elderly living in residential health care; (**c**) Comoobility share, i.e., number of positive cases with comorbidities divided by the total number of positive cases; (**d**) Symptomatic to asymptomatic ratio, i.e., number of symptomatic positive cases divided by the asymptomatic number of cases.

**Figure 3 ijerph-19-05921-f003:**
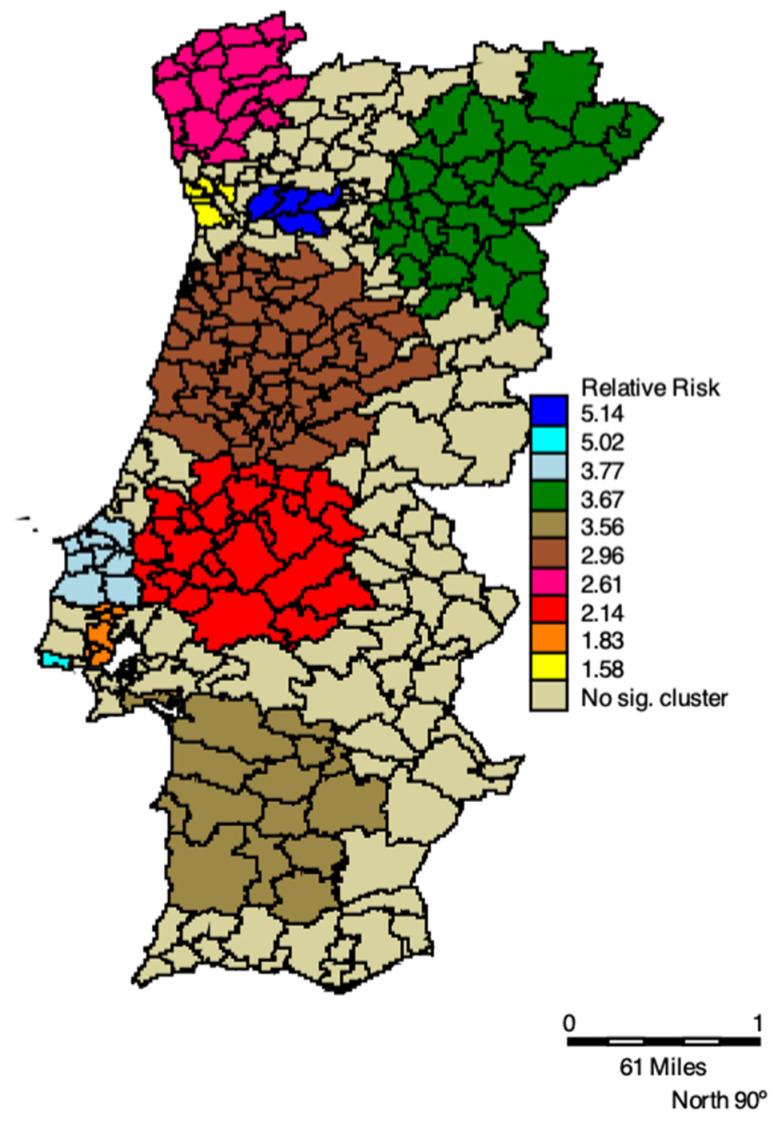
Space-time clusters of the elderly living in Residential care homes between 3 March and 31 December 2020.

**Figure 4 ijerph-19-05921-f004:**
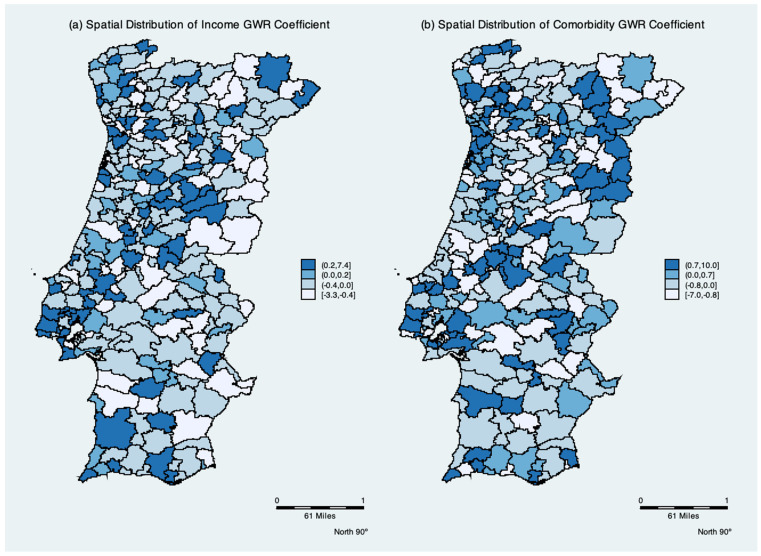
Spatial distribution of GWR parameter estimates. Subfigure (**a**) Spatial distribution of the income GWR parameter; (**b**) Spatial distribution of the comrbidity GWR parameter.

**Table 1 ijerph-19-05921-t001:** Total Positive Cases, Mean of Positive Cases and Proportion Positive by 1000 patients.

	Mar.	April	May	June	July	Aug.	Sep.	Oct.	Nov.	Dec.
Total Positive cases in Portugal (No.)	6648.00	13,960.00	6865.00	8312.00	7596.00	5709.00	12,834.00	34,776.00	62,002.00	49,136.00
Total Positive cases in elderly living in RCH (No.)	656.00	1798.00	500.00	426.00	350.00	329.00	737.00	1917.00	3819.00	3549.00
Average elderly positive cases per day (No.)	22.62	59.93	16.13	14.20	11.29	10.61	24.57	61.84	127.30	114.48
Average elderly Proportion Positive per day (No.)	59.92	125.44	75.07	50.46	44.51	51.68	59.37	55.79	63.87	73.88

**Table 2 ijerph-19-05921-t002:** Portuguese municipalities in space-time clusters between 3 March and 31 December 2020 and respective relative risks of COVID-19 infection among the elderly living in residential care homes.

Outbreak Date/End Date	No. Days in Signal	No. Municipality	Radius (km)	Observed Cases	Expected Cases	Relative Cases	Relative Risk (RR)	Test Statistic	Color in Figure 3
30 Sep./13 Dec.	75	29	74	479	133.85	3.58	3.67	269.873 ***	Green
27 Mar./23 Apr.	28	51	63	453	156.57	2.89	2.96	188.019 ***	Brown
24 Mar./13 Apr.	21	19	54	363	141.44	2.57	2.61	122.346 ***	Pink
25 Oct./30 Dec.	67	27	51	411	194.95	2.11	2.14	92.188 ***	Red
15 Oct./02 Nov.	19	15	70	164	46.4	3.53	3.56	89.95 ***	Light Brown
27 Jun./03 Jul.	7	1	0	76	15.2	5	5.02	61.666 ***	Turquoise
29 Mar./28 May	61	5	12	538	346.3	1.55	1.58	46.664 ***	Yellow
29 Nov./7 Dec.	9	6	15	236	129.74	1.82	1.83	35.343 ***	Orange
12 Dec./18 Dec.	7	8	26	56	14.88	3.76	3.77	33.148 ***	Light Blue
12 Jun./18 Jun.	7	4	14	24	4.67	5.14	5.14	19.963 ***	Navy Blue

Note: *** statistically significant (*p* < 0.001).

**Table 3 ijerph-19-05921-t003:** Geographically Weighted Regression Estimation of Elderly COVID-19 Relative Risk in Portuguese municipalities between 3 March and 31 December 2020.

	OLS	GWR			
		Si	Mean	Min	Max
Income	−0.215 ** (0.089)	0.978 ^+++^	−0.036	−3.300	7.400
Comorbidity	0.503 (0.389)	1.224 ^+++^	−0.125	−7.000	10.000
Constant	2.744 *** (0.513)	2.218 ^+++^	2.044	−28.090	17.104
Test Statistics					
Moran’s I	0.271 ***				
Bandwidth		0.257 ^+++^			
F(2275)	3.54 **				
−2(log likelihood)	1032.268				
AIC	1038.268				
N	278	278	278	278	278

Monte Carlo tests of spatial variability: ^+++^ Rejects the null at the 1% level. T-tests: ** Rejects the null at the 5% level. *** Rejects the null at the 1% level. Standard Errors in parentheses. Data Source: SINAVE and INE.

## Data Availability

Since our analysis is of public health data at the municipality level (with no individual or identifiable patient data), separate research ethics review was not needed, and this work was undertaken under generic approval by the Portuguese general health directorate (DGS) and by the SNS and ISCTE-IUL Dataloch partnership agreement. The data are provided by the Business Intelligence of the information technology support platform for the National Epidemiological Surveillance System (BI SINAVE). The study was exempted from submitting a report to the Research Ethics Committee; nevertheless, the researchers used caution and respected the ethical standards pertaining to research.

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
