# Peer review of "Spatial and Temporal Analysis of COVID-19 in the Elderly Living in Residential Care Homes in Portugal"

_ijerph, 2022, doi:10.3390/ijerph19105921_

Round 1
Reviewer 1 Report
Congratulations to the author for her excellent and complete work. Some comments “in favor” of improving the version of the submitted manuscript are made..- Tittle. Perhaps add at the end of tittle "in Portugal".
.- Abstract. Introduction section is missing. Conclusion, perhaps an additional sentence could be added (based on the information provided in the Conclusions part of the manuscript)
.- Keywords. ok.
.- Introduction. Consider shortening/abbreviating some paragraph. Perhaps the first and/or third and fourth paragraphs could be shortened. In relation to the references to KDE, the Kulldorff scan method and GWR, they could be ignored, indicating that they will be explained in the methodology section. Only the introduction includes 31 bibliographic citations. The rest of the manuscript includes 19 citations, this is less than half of the total citations.
When you say “between 3 March and 31 December 2020” it is better to say “between March 3 and December 31, 2020”
.- Methods. Consider shortening/abbreviating some paragraph.
..- Results. Consider shortening/abbreviating some paragraph. Perhaps the information provided in Figure 2 could be summarized.
.- Discussion. Consider shortening/abbreviating some paragraph. When in the first paragraph it says “and in several European countries” perhaps it should say “as it happens in several European countries”.
In the third paragraph, he again explains results by references to Figure 2. This comments are not applicable in this section.
In the same paragraph, when it says “The average density surrounding the main urban areas can indicate flows of travelers and flows of people who work in the city and then return to their residences in the surroundings”. Maybe it could be removed.
.- Conclusions. The sentences in this section are excessively long. I would be interested in redoing or dividing a long sentence into two phrases.
.- References. Of the 50 citations, more than 75% are recents (this is less than 5 years). Congratulations!.
.- Tables and Figures. 5 Figures and 3 Tables. Even being very interesting and complete, could it be considered to eliminate someone? Maybe Figure 3?
NOTE: Avoid excessive repetitions of terms or expressions, e.g. “RCH”, 31 times in the manuscript. Only in paragraph 3 of the Method section “RCH” is repeated 6 times; “March 3 to December 31, 2020” is repeated 10 times and the expression “March to December 2020”, 4 times. The acronym “OLS” appears 4 times in the paragraph describing the findings in Table 3. It is recommended to search for synonyms or similar expressions.
Author Response
Thank you for the opportunity to revise this paper. We are very grateful for the thorough and constructive remarks of the reviewers. We have endeavored to comprehensively address the points raised by the reviewers and we feel that our paper has been greatly enhanced as a result.
If reviewers agree, we would like to thank the anonymous reviewers in the Acknowledgments section for critically reading the manuscript and suggesting substantial improvements.
Highlighted in yellow in the manuscript are the revisions according with the comments and suggestions.
Reviewer 1
Congratulations to the author for her excellent and complete work. Some comments “in favor” of improving the version of the submitted manuscript are made.
.- Tittle. Perhaps add at the end of tittle "in Portugal".
R: Thank you for this recommendation. We add at the end of tittle "in Portugal".
.- Abstract. Introduction section is missing. Conclusion, perhaps an additional sentence could be added (based on the information provided in the Conclusions part of the manuscript)
R: Thank you for this recommendation. We add an additional sentence highlighted in yellow.
.- Keywords. ok.
.- Introduction. Consider shortening/abbreviating some paragraph. Perhaps the first and/or third and fourth paragraphs could be shortened. In relation to the references to KDE, the Kulldorff scan method and GWR, they could be ignored, indicating that they will be explained in the methodology section. Only the introduction includes 31 bibliographic citations. The rest of the manuscript includes 19 citations, this is less than half of the total citations.
R: Thank you for this recommendation. We add shorten/abbreviate the paragraphs highlighted in yellow (1st, 3rd). the 4th paragraph was deleted as suggested by reviewer 3. Consequently, we removed the references of KDE, Kulldorff scan method and GWR.
When you say “between 3 March and 31 December 2020” it is better to say “between March 3 and December 31, 2020”
R: Thank you for this recommendation. We wrote “between March 3 and December 31, 2020”.
.- Methods. Consider shortening/abbreviating some paragraph.
R: Thank you for this recommendation. We tried to shorten/abbreviate the paragraphs highlighted in yellow.
..- Results. Consider shortening/abbreviating some paragraph. Perhaps the information provided in Figure 2 could be summarized.
R: Thank you for this recommendation. We tried to shorten/abbreviate the paragraphs highlighted in yellow. The information provided in Figure 2 was summarized.
.- Discussion. Consider shortening/abbreviating some paragraph. When in the first paragraph it says “and in several European countries” perhaps it should say “as it happens in several European countries”.
R: Thank you for this recommendation. We wrote “as it happens in several European countries”. We tried to shorten/abbreviate the paragraphs highlighted in yellow
In the third paragraph, he again explains results by references to Figure 2. This comments are not applicable in this section.
R: Thank you for this recommendation. We deleted these comments.
In the same paragraph, when it says “The average density surrounding the main urban areas can indicate flows of travelers and flows of people who work in the city and then return to their residences in the surroundings”. Maybe it could be removed.
R: Thank you for this recommendation. We deleted “The average density surrounding the main urban areas can indicate flows of travelers and flows of people who work in the city and then return to their residences in the surroundings”.
.- Conclusions. The sentences in this section are excessively long. I would be interested in redoing or dividing a long sentence into two phrases.
R: Thank you for this recommendation. We try to shorten the sentences.
.- References. Of the 50 citations, more than 75% are recents (this is less than 5 years). Congratulations!.
R: Thank you so much.
.- Tables and Figures. 5 Figures and 3 Tables. Even being very interesting and complete, could it be considered to eliminate someone? Maybe Figure 3?
R: Thank you for this recommendation. Figure 3 was eliminated.
NOTE: Avoid excessive repetitions of terms or expressions, e.g. “RCH”, 31 times in the manuscript. Only in paragraph 3 of the Method section “RCH” is repeated 6 times; “March 3 to December 31, 2020” is repeated 10 times and the expression “March to December 2020”, 4 times. The acronym “OLS” appears 4 times in the paragraph describing the findings in Table 3. It is recommended to search for synonyms or similar expressions.
R: Thank you for this recommendation. We tried to avoid repetitions.
Reviewer 2 Report
The authors analyse a highly relevant issue of the pandemic, the elderly persons infected. From the the statiscal point of view, I was not able to check the methods, since the documentation of the analysis is very poor. The authors do not document, what software they are using. In my view the analysis should be documented such that is reproducabel (i.e. the code and the data should be provided)
Details:
1) In the GWR - regresssion It is not clear what variables are used. There is no model equation. The authors do not say, what kind of software they use.
2) The use of Kulldorf scan method is also not ducumented
3) The overall conclusion
"Then, analyzing locally these high-risk clusters with income and comorbidities data, the results suggest improving both income and health levels in the southwest of Portugal, in the environs of large cities like Lisbon and Porto, and in the northwest of Portugal, where an increased risk of infection of elderly due to covid-19 was detected."
is not in relationship to the paper about covid ?
3) The tables and the graphics are problematic. A decent caption is missing.
in Fig one the scale for one curve is missing.
4)I do not understand the sentence: where is the difference between elderl linving in RCH and elderlxyliving in residential car homes ?
"The elderly living in RCH accounted for approximately 7% (14,081 cases) of the total positive cases. Using the number of positive cases of elderly living in residential care homes per 1000 total positive cases, it represented around 10% (20,041 cases) of the positive cases."
Author Response
Thank you for the opportunity to revise this paper. We are very grateful for the thorough and constructive remarks of the reviewers. We have endeavored to comprehensively address the points raised by the reviewers and we feel that our paper has been greatly enhanced as a result.
If reviewers agree, we would like to thank the anonymous reviewers in the Acknowledgments section for critically reading the manuscript and suggesting substantial improvements.
Highlighted in yellow in the manuscript are the revisions according with the comments and suggestions.
Reviewer 2
The authors analyse a highly relevant issue of the pandemic, the elderly persons infected. From the the statiscal point of view, I was not able to check the methods, since the documentation of the analysis is very poor. The authors do not document, what software they are using. In my view the analysis should be documented such that is reproducabel (i.e. the code and the data should be provided)
Details:
1) In the GWR - regresssion It is not clear what variables are used. There is no model equation. The authors do not say, what kind of software they use.
R: Thank you for this recommendation. We included the equation in section 2.3, as well as the software used, highlighted in yellow.
2) The use of Kulldorf scan method is also not ducumented
R: Thank you for this recommendation. We included the software in section 2.2, highlighted in yellow.
3) The overall conclusion
"Then, analyzing locally these high-risk clusters with income and comorbidities data, the results suggest improving both income and health levels in the southwest of Portugal, in the environs of large cities like Lisbon and Porto, and in the northwest of Portugal, where an increased risk of infection of elderly due to covid-19 was detected."
is not in relationship to the paper about covid ?
R: Thank you for this recommendation. We rewrote this conclusion to make it clearer.
3) The tables and the graphics are problematic. A decent caption is missing.
in Fig one the scale for one curve is missing.
R: Thank you for this recommendation. We included the caption missing in Figure 1. We also try to improve the other captions, highlighted in yellow.
4)I do not understand the sentence: where is the difference between elderly living in RCH and elderly living in residential car homes ?
"The elderly living in RCH accounted for approximately 7% (14,081 cases) of the total positive cases. Using the number of positive cases of elderly living in residential care homes per 1000 total positive cases, it represented around 10% (20,041 cases) of the positive cases."
R: Thank you for this recommendation. We rewrote this sentence in order to make it clear the difference between the two measures.
Reviewer 3 Report
Manuscript ID: ijerph-1684350
Spatial and Temporal Analysis of COVID-19 in the Elderly living in Residential Care Homes
Method: The strength of the chosen spatial and temporal analysis should be highlighted.
Line 80-93: Remove methodological sentences, and move them into the method section.
Line 94-97: Remove this paragraph
Discussion: Add discussion about the worrying confirmed cases in the highly affected RCHs. What have been implemented so far, the constraints, and immediate action plan for those affected older peoples. This definitely reflect the public health aspects in this topic.
Conclusion: No future direction is projected in this paragraph
Format: The reference citation order in the text needs further revision.
Reference list: The first alphabet for each reference is missing.
Author Response
Thank you for the opportunity to revise this paper. We are very grateful for the thorough and constructive remarks of the reviewers. We have endeavored to comprehensively address the points raised by the reviewers and we feel that our paper has been greatly enhanced as a result.
If reviewers agree, we would like to thank the anonymous reviewers in the Acknowledgments section for critically reading the manuscript and suggesting substantial improvements.
Highlighted in yellow in the manuscript are the revisions according with the comments and suggestions.
Reviewer 3
Method: The strength of the chosen spatial and temporal analysis should be highlighted.
R: Thank you for this recommendation. We included a paragraph to subsection 2.2 explaining the strength of spatial temporal analysis, highlighted in yellow.
Line 80-93: Remove methodological sentences, and move them into the method section.
R: Thank you for this recommendation. We Removed this paragraph.
Line 94-97: Remove this paragraph
R: Thank you for this recommendation. We Removed this paragraph.
Discussion: Add discussion about the worrying confirmed cases in the highly affected RCHs. What have been implemented so far, the constraints, and immediate action plan for those affected older peoples. This definitely reflect the public health aspects in this topic.
R: Thank you for this recommendation. We wrote the last paragraph of section 4 in order to highlight some public health measures.
Conclusion: No future direction is projected in this paragraph
R: Thank you for this recommendation. We included future research in the conclusion section.
Format: The reference citation order in the text needs further revision.
R: Thank you for this recommendation. We revised the order of the reference list.
Reference list: The first alphabet for each reference is missing.
R: Thank you for this recommendation. We revised the format of the reference list in order to show the first alphabet.
Round 2
Reviewer 2 Report
Most of my comments are answered ok .
The authors dis not answer to my point of reproducable research. (Provide code and Data)
Figure 2 is still of terrible quality. There should be some units given for the categories.
The results KDE is nor documented
Reviewer 3 Report
The author has successfully responded to my comments. The current version is satisfactory and acceptable for publication.
This manuscript is a resubmission of an earlier submission. The following is a list of the peer review reports and author responses from that submission.